# Promising Effects of 3-Month Period of Quercetin Phytosome^®^ Supplementation in the Prevention of Symptomatic COVID-19 Disease in Healthcare Workers: A Pilot Study

**DOI:** 10.3390/life12010066

**Published:** 2022-01-04

**Authors:** Mariangela Rondanelli, Simone Perna, Clara Gasparri, Giovanna Petrangolini, Pietro Allegrini, Alessandro Cavioni, Milena Anna Faliva, Francesca Mansueto, Zaira Patelli, Gabriella Peroni, Alice Tartara, Antonella Riva

**Affiliations:** 1IRCCS Mondino Foundation, 27100 Pavia, Italy; mariangela.rondanelli@unipv.it; 2Department of Public Health, Experimental and Forensic Medicine, University of Pavia, 27100 Pavia, Italy; 3Department of Biology, College of Science, Sakhir Campus, University of Bahrain, Zallaq 32038, Bahrain; simoneperna@hotmail.it; 4Endocrinology and Nutrition Unit, Azienda di Servizi alla Persona “Istituto Santa Margherita”, University of Pavia, 27100 Pavia, Italy; clara.gasparri01@universitadipavia.it (C.G.); alessandro.cavioni01@universitadipavia.it (A.C.); milena.faliva@gmail.com (M.A.F.); francesca.mansueto01@universitadipavia.it (F.M.); zaira.patelli01@universitadipavia.it (Z.P.); alice.tartara01@universitadipavia.it (A.T.); 5Research and Development Unit, Indena SpA, 20139 Milan, Italy; giovanna.petrangolini@indena.com (G.P.); pietro.allegrini@indena.com (P.A.); antonella.riva@indena.com (A.R.)

**Keywords:** quercetin, COVID-19, SARS-CoV-2, prevention

## Abstract

Quercetin, for its crucial properties, fulfills the need for a multifactor action that is useful for the potential counterbalance of a COVID-19 infection. Given this background, the aim of the study was to evaluate the potential effect of 3 months’ supplementation with Quercetin Phytosome^®^ (250 mg twice a day) as prevention against symptomatic COVID-19. In total, 120 subjects were enrolled (males, 63; females, 57; age 49 ± 12), with 60 in the supplementation group and 60 in the placebo group. No significant differences were detected between groups in terms of gender, smoking, and chronic disease. Subjects underwent rapid COVID-19 diagnostic tests every 3 weeks. During our study, 5 subjects had COVID-19, 1 out of 60 subjects in the quercetin group and 4 out of 60 in the control group. Complete clinical remission was recorded at 7 and 15 days in the quercetin and placebo groups, respectively. Analysis showed that, at 5 months, the COVID free survival function (risk of infection) was 99.8% in subjects under quercetin supplementation and 96.5% in control group. As shown by the value of EXP(B), those who had taken the supplement had a protection factor of 14% more to not contract the COVID-19 infection than that of those who had taken a placebo. Obtained results are encouraging, but further studies are required to add quercetin as regular prophylaxis.

## 1. Introduction

The viral infection, which began in December 2019 and was declared a pandemic by the World Health Organization (WHO) on 11 March 2020, has forced humanity to face a global health crisis. It is caused by a new coronavirus, SARS-CoV-2, which has led to a dramatic number of global infections and deaths (4.9 million) with severe acute respiratory syndromes [1].

A recent article by Lee [2] and previously a review by Shi [3] reported that symptomatic COVID-19 infection is associated with a first phase with prevalent immune involvement, and a second phase characterized by a cytokine storm and macrophage activation syndrome, showing how in serum samples taken from COVID-19 patients, senescence-associated secretory phenotypes (SASP), such as interleukin-8 (IL-8), plasminogen activator inhibitor-1 (PAI-1), metalloproteinase chemokine ligand 2 (CCL2), metalloproteinase 1 (MMP-1), metalloproteinase 9 (MMP-9), and tissue metallopeptidase inhibitor 1 (TIMP-1), were overexpressed factors during symptomatic infection.

This cytokine storm (the release of large amounts of proinflammatory cytokines, interferons alfa (IFN-a), interferons gamma (IFN-g), interleukin 1b (IL-1b), interleukin 6 (IL-6), interleukin 12 (IL-12), interleukin 18 (IL-18)) and uncontrolled inflammation are major mechanisms for acute respiratory distress syndrome (ARDS) and multiple organ failure [4,5,6].

Moreover, COVID-19 appears to be closely linked to senescence-immune escalation with the cytokine storm, thrombotic events, and the abnormal activation of macrophages and immune system as a consequence [2].

According to Lee and Shi [2,3], therapies should try to increase the immune response during the first phase, and suppress it in the second phase and counterbalance inflammatory activity; furthermore, in both phases, senolytic activity is required. Given the complexity of the organism’s response to COVID-19 explained in the 2 reviews by Lee and Shi [2,3], it is clear why no targeted drugs have been identified in either the preventive or therapeutic field [7], as multiple biological activity is required to fight the virus. The scientific literature has reported that some botanicals possess this multibiological activity, as they have the ability to modulate both immune and inflammatory responses in individuals suffering from respiratory symptoms [8].

Among these botanicals, quercetin is supported by an extensive scientific literature in vitro and in animal models, as reported in a recent review by Aucoin that demonstrated its capacity to modulate multiple biological pathways, and acknowledged 4 major properties helpful for respiratory and immune health: antioxidant, anti-inflammatory, immunomodulating, and antiviral [9]. So, quercetin fulfills the need for multifactorial action potential that is useful for counterbalancing a COVID-19 infection. Due to its antioxidant and antimicrobial effects, quercetin has the capacity of inhibiting key proteins involved in the coronavirus’ infective cycle [10]. Riva and colleagues [11] showed scientific evidence of positive effects due to the high antioxidant capacity of Quercetin Phytosome^®^. In triathletes, the supplementation of Quercetin Phytosome^®^ was more effective than the control (no treatment) in reducing oxidative stress (by measurement plasma free radicals). Moreover, a second human study [12] confirmed the positive antioxidant effects of Quercetin Phytosome^®^ after supplementation in individuals suffering from allergic rhinitis and asthma (airway allergy), being more effective than the standard management in reducing oxidative stress (by measurement of plasma free radicals).

Considering the anti-inflammatory activity of quercetin, functional similarities were reported between dexamethasone and quercetin, with both being permeability-glycoprotein (PGP) inducers. PGP, a membrane transporter, helps in the retention of various drugs, but proinflammatory cytokines released during infections inhibit its expression and activity. As PGP inducers, both dexamethasone and quercetin are reported to possibly inhibit the cytokine-storm-like consequences in COVID-19 patients [13].

Furthermore, quercetin possesses inhibitory activity on viral proteases, which plays essential roles in viral replication; specifically, 6LU7 was evaluated as the main protease (Mpro) found in COVID-19. Quercetin resulted in the formation of H-bonds with some 6LU7 amino acids (SARS-CoV-2 principal protease) located in the COVID-19 Mpro active site [14,15].

Furthermore, quercetin has additional activities specifically aimed at counteracting COVID-19. Quercetin alters the expression of 98 of the 332 human genes (30%) that encode SARS-CoV-2 protein targets, thus potentially interfering with the functions of 23 of 27 (85%) of the viral proteins of SARS-CoV-2 in human cells [16].

Lastly, quercetin was identified between the top scoring ligands for viral spike protein, the angiotensin-converting enzyme 2 (ACE2) receptor interface, thus potentially being able to limit the viral recognition of host cells and/or disrupt host–virus interactions [17].

A recent study in animal models demonstrated that quercetin flavonoids, in combination with dasanitinib, allowed for a substantial reduction in senescent cells in COVID-19-infected animals’ respiratory tract and a dramatic reduction in SASP cytokines in their blood serum.

Quercetin addresses the highest number of cellular targets compared with other senolytic agents.

With regard to human studies, a study on effectiveness in prevention, and 2 studies on efficacy in the course of COVID-19 infection have been published.

The study on prevention revealed that 500 mg of quercetin supplementation, in association with 500 mg of vitamin C and 50 mg of bromelain, was significantly protective to healthcare workers in a 3-month period: 1 out of 71 healthcare workers using quercetin and 9 out of 42 healthcare workers in the control group had COVID-19 [18].

As regards studies on efficacy during infection, the pilot studies carried out on human subjects by Di Pierro and collaborators were significant: a first study conducted on a sample of 42 subjects that had tested positive to the molecular test for SARS-CoV-2, not in serious condition, and randomly divided into two treatment groups showed that patients treated with standard therapy + Quercetin Phytosome^®^ supplementation (for 14 days, the first 7 with 600 mg and last 7 with 400 mg) had a lower temporal persistence of infection (faster negativization of the antigen test) and a more rapid resolution of symptoms than those in the group treated with standard therapy alone [19]; a second study conducted on a larger sample (152 subjects affected by COVID-19 and not in serious condition), randomly divided into two groups of 76, found that subjects treated with standard therapy + Quercetin Phytosome^®^ (400 mg/day for 30 days) had better clinical outcomes (lower hospitalization rate, shorter hospitalization time, reduced need for oxygen therapy) than those treated with standard therapy alone [20].

Figure 1 shows the pleiotropic actions of quercetin in COVID-19.

Given this background, the aim of the present pilot study was to evaluate if a 3-month period of quercetin supplementation (500 mg of Quercetin Phytosome^®^) is useful in prevention of COVID-19 infection and against symptomatic COVID-19 disease in healthcare workers.

## 2. Materials and Methods

This was a single-center, prospective, randomized, controlled cohort study.

### 2.1. Participants

A total of 120 healthcare workers were enrolled and signed the informed consent. Adult subjects aged 20–60 years old were included. Exclusion criteria were: previous infection with SARS-CoV-2 (positive SARS-CoV-2 PCR or IgG serology), pregnancy or lactation, any kind of contraindication to quercetin, and evidence of unstable systemic disease.

The Ethics Committee of the University of Pavia approved the study (Ethics Committee approval number: 1222/01022021). This study was registered on 7 September 2020 at ClinicalTrials.gov with number NCT05037240.

Demographic features of the participants, such as age, gender, smoking, physical activity, chronic disease, and medication were all recorded.

Subjects underwent rapid COVID-19 diagnostic tests every 3 weeks. In the event that the subject tested positive for COVID-19 in the rapid diagnostic tests, they were subjected to a confirmative SARS-CoV-2 reverse-transcription polymerase chain reaction (RT-PCR) test through nasal and pharyngeal swab samples.

In the case of positivity to COVID-19, in order to recognize a patient’s clinical deterioration, the National Early-Warning Score (NEWS) was applied [21,22].

NEWS (0–20, higher = worse) comprises seven physiological variables (respiratory rate, oxygen saturation, supplemental oxygen, temperature, systolic blood pressure, heart rate, and level of consciousness) that are often integrated in early-warning systems to identify high-risk patients in acute care settings.

A maximal follow-up period was determined to be at 3 months. The termination of the participant’s use of the quercetin supplement earlier than 3 months or having an active coronavirus infection was considered to be the primary endpoint.

Figure 2 represents the flowchart of the study.

At the end of the follow-up period, COVID-19 rapid diagnostic tests (RDT) were used to detect COVID-19 IgG and M positivity in all cases.

### 2.2. Dietary Supplement

In the intervention group, named Quercetin Group (QG), the supplementation of 500 mg of Quercetin Phytosome^®^ was initiated daily in 2 divided doses for 60 subjects, whereas the 60 other subjects were supplemented with a placebo (control group). Quercetin Phytosome^®^ were formulated as white film-coated tablets. Each film-coated tablet contained 250 mg of Quercetin Phytosome^®^ and food-grade ingredients. In order to guarantee the fast bioaccessibility of Quercetin Phytosome^®^, film-coated tablets were characterized by fast disintegration time (<30 min). The placebo was formulated as white film-coated tablets having the same appearance as that of Quercetin Phytosome^®^ tablets.

All subjects took measures to contain the spread of the SARS-CoV-2 virus (wearing masks, social distancing, hand disinfection, etc.)

### 2.3. Statistical Analysis

Clinical data are presented as means ± SD. The supplementation and placebo groups were compared with the Student’s test for numerical variables, and Fisher’s test for categorical variables. The sample size of 120 subjects was calculated on the basis of a previous study by Arslan et al. [18].

SPSS version 21.0 (SPSS, Chicago, IL, USA) was used for statistical analysis. Statistical significance was reached when the probability (*p*) value was <0.05, and changes were referred to as significant at this p value. For the primary outcome, the hazard ratio (HR) of transmission risk was evaluated by Cox’s regression method to estimate the infection contagion rate ratio.

## 3. Results

As shown in Table 1, a total number of 120 subjects aged 49.29 ± 12.94 years was included, 60 in the supplementation group and 60 in the placebo group.

The mean age of QG group was 50.88 ± 12.08 years, and the mean age of the control group was 47.70 ± 13.67 years. No significant differences were detected between groups in terms of gender, smoking, and the presence of chronic disease at baseline.

Table 2 reports the demographic characteristics of the sample: 52.5% were males, the majority were not smokers (82.5%), workers in contact with other people (64%), and with a level of education of high school (46%) or university degree (37%).

The primary endpoint was time to clinical improvement up to day 17 of the infection.

As shown in Figure 2, a total of 5 subjects had COVID-19, 1 out of 60 subjects administered with quercetin supplementation, and 4 out of 60 in the control group. Four subjects allocated into the placebo group showed lower clinical improvement compared to the one subject in treatment with Quercetin Phytosome^®^ on the basis of the assessment of the mean score of NEWS scale (the difference between groups was recorded at day 7, when the patient in the intervention group had a score of 0 as the level of symptoms).

Complete clinical improvement in the quercetin group was recorded at 7 and 15 days in the placebo group, as shown in Figure 3. None of the subjects required hospitalization.

Table 3 shows the healing status regarding 5 positive cases of COVID-19. The subject allocated in the supplementation group was negative at 10 days, while the 4 subjects in the placebo group had a negative swab after 10 + 7 days.

Figure 4 shows that, at 5 months, the COVID free survival function (avoiding infection) was 99.8% in subjects under quercetin supplementation and 96.5% in control group.

The protective potential of quercetin against COVID-19 was investigated by using the Cox proportional-risk model in subjects receiving quercetin supplementation versus the control group, as shown in Table 4. The risk of having COVID-19 in the placebo versus the quercetin group was HR Exp(B) = 14.09 (CI 95% 1.094; 181.596, *p* = 0.042) A hazard ratio of 14.04 means that subjects without quercetin support were at 14 times the risk of being infected with COVID 19 compared to people with quercetin support.

There was no statistically significant difference between the quercetin and control groups in terms of gender, smoking, antihypertensive medication use, and presence of chronic disease.

## 4. Discussion

The present pilot study revealed that quercetin supplementation was significantly protective against symptomatic coronavirus infection in a 3-month period. A total number of 5 subjects had COVID-19, 1 out of 60 subjects supplemented with Quercetin Phytosome^®^ and 4 out of 60 in the control group. Moreover, the four subjects allocated into the placebo group showed lower clinical improvement, assessed by NEWS, compared to the one patient in the quercetin group.

These results are in accordance with the observations by Arslan et al., who revealed that quercetin supplementation was significantly protective for healthcare workers in a 3-month period. One healthcare worker in the intervention group and 9 out of 42 in the control group had COVID-19. However, the supplement contained not only 500 mg of quercetin, but also 500 mg of vitamin C and 50 mg of bromelain [18].

Our analysis revealed that complete clinical improvement in the quercetin group was recorded at 7 days, and at 17 days in the placebo group. In fact, the subject in the intervention group was negative at 10 days, while the 4 subjects in the placebo group had a negative swab after 10 + 7 days. None of the patients required hospitalization.

These data are in line with the results obtained by Di Pierro et al., who demonstrated that Quercetin Phytosome^®^ statistically shortened the timing of molecular test conversion from positive to negative, reducing at the same time symptom severity and negative predictors of COVID-19 [19].

A prospective, randomized, controlled, and open-label study demonstrated that the administration of a daily dose of Quercetin Phytosome^®^ for 30 days in 152 COVID-19 outpatients resulted in a reduction in the frequency and length of hospitalization, the need for noninvasive oxygen therapy, progression to intensive care units, and in the number of deaths. In combination with standard care, when used in the early stages of viral infection, quercetin could improve the early symptoms and help in preventing the severity of progression of COVID-19. In addition, the authors suggested possible antifatigue and proappetite properties of quercetin [20].

Moreover, our analysis showed that, at 5 months, the COVID-19 free survival function (risk of infection) was 99.8% in subjects under quercetin supplementation, and 96.5% in the control group. As shown by the value of EXP(B), those who had taken the supplement had a protection factor of 14% more to not contract COVID-19 infection than that of those who had taken the placebo.

As the COVID-19 pandemic is a recent global health problem, there are currently only a few other human clinical studies about quercetin supplementation in the literature, and in these clinical studies, subjects were administered multicomponent supplements and not quercetin alone [18,23].

Polyphenols are the largest class of natural bioactive compounds, categorized as flavonoids (for example, quercetin) and nonflavonoids, and these compounds have been described as effective antiviral agents. This is because they can inhibit coronavirus enzymes, blocking replication and infection [24].

To date, there is a considerable amount of data describing the safe profile and the potential antiviral, anti-inflammatory, and thrombin-inhibitory actions of quercetin [25,26].

A recent article by Lee [2] and a previous review by Shi [3] reported that symptomatic COVID-19 infection is associated with a first phase of prevalent immune involvement. Subsequently, a second phase is characterized by a cytokine storm and macrophage activation syndrome.

We, therefore, suppose that quercetin is involved in the first phase, activating the immune system, and counteracting cytokines storm and senescent cells. Recent data support the use of quercetin, and in particular its formulation in the phytosome, as a promising ingredient for the mitigation of COVID-19 manifestations. New scientific papers have been published adding evidence that reinforce this remarkable potential asset, such as a study where quercetin was reported to be a mitigation agent for COVID-19 on the basis of genomic analysis in human cells [16]. Moreover, the Lee paper cited above reported Quercetin Phytosome^®^ as possible senolytic agent that is suitable for the management of COVID-19 [2]. Briefly, senescent cells can be considered to be a therapeutic target in COVID-19, whose early elimination might mitigate the course of the disease. Known senolytic compounds (i.e., mavitoclax and the combination of dasatinib + quercetin) were tested in SARS-CoV-2-infected animals using only the solvent as placebo control. Animals with senolytic interventions presented a substantial reduction in senescent cells in their respiratory tract and a dramatic reduction in SASP cytokines in blood serum. Additionally, the navitoclax group showed high levels of side effects; on the other hand, the dasatinib + quercetin group did not show any adverse event. Lee’s paper also analyzed results from two recent human studies on Quercetin Phytosome^®^ [19,20] confirming the potential positive effect of quercetin-based senolytic intervention, able to reduce many negative outcomes including length of hospitalization, oxygen therapy, referral to intensive care units, and number of deaths. Lee and colleagues concluded that quercetin could attenuate COVID-19 lung disorders and systemic inflammation during active infection, and even mitigate the chronic postinfection impairment known as long COVID due to its senolytic activity [2].

The main limitation of this study was the lack of the evaluation of immunity response and cytokine production, which would have been useful in correlating clinical efficacy with the immune and cytokine response, demonstrating the multibiological activity of quercetin theorized by Lee et al. [2], and Shi et al. [3]. Moreover, another limitation of this study regards its small sample size and short duration of the intervention, which was limited to three months. Lastly, another limitation was the enrolled subjects, who were only healthcare workers, and this may limit generalization.

Regarding the quercetin formulation that was chosen for the study, for the optimization of oral bioavailability, the supplementation was 500 mg of Quercetin Phytosome^®^. Quercetin Phytosome^®^ is a 100% food-grade delivery system of quercetin. Quercetin from *Sophora japonica* L. was formulated with Phytosome^®^ technology to optimize its oral bioavailability as demonstrated in a pharmacokinetic study in healthy volunteers [27]. Phytosome delivery system is a solid dispersion that combines botanicals or natural compounds into a 100% food-grade sunflower lecithin matrix, amphipathic molecules that act as inhibitors of self-aggregation, and effective wetting agents that are able to maintain the original components’ phytochemical profile while improving the health benefits.

There is much evidence on the potential favorable biological use of Quercetin Phytosome^®^. First, Quercetin Phytosome^®^ enhances the plasmatic levels of quercetin and improves quercetin bioabsorption by up to 20 times after the oral administration of a single dose vs. unformulated quercetin [27] in a pharmacokinetic study in humans. Furthermore, Quercetin Phytosome^®^ has a remarkably favorable safety profile [28]. The interaction between Quercetin Phytosome^®^ and the human microbiota was also elucidated [29]. This last piece of evidence sustained that the Quercetin Phytosome^®^ formulation was more stable than unformulated quercetin after interaction with the intestinal microbiota. Indeed, the phytosome can slow down the intestinal microbial degradation of quercetin, allowing for more time and the better dispersion of the single molecule to be absorbed, thus overcoming one of the possible reasons for quercetin’s poor oral bioavailability as reported by Riva and colleagues [27].

## 5. Conclusions

The results obtained in the present pilot study are innovative and encouraging, but further studies including a large number of participants and with a longer follow-up are required to consider quercetin as regular prophylaxis.

The global emergency due to SARS-CoV-2 is critical; thus, guidelines provided by international health authorities, local governments, and ministries of health must be followed. Social distancing, avoiding contact, and consulting a doctor if any symptoms occur are strict and necessary rules that everybody must follow to ward off contagion. In such a scenario, some active botanical principles have the potential capacity to modulate the aggressiveness of the infection, such as quercetin supplemented in a high-performance formulation.

## Figures and Tables

**Figure 1 life-12-00066-f001:**
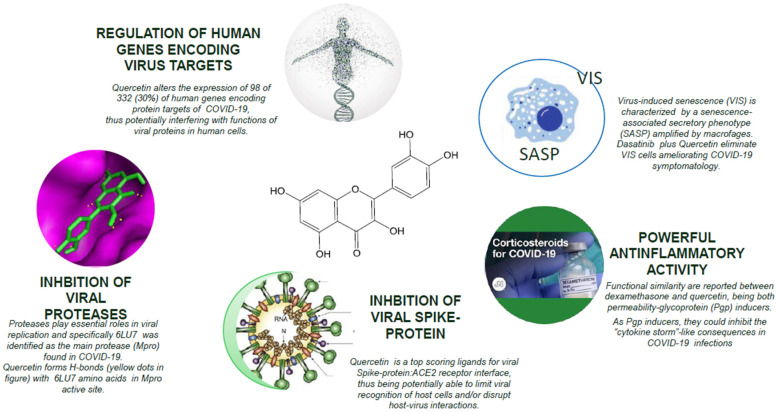
Pleiotropic actions of quercetin in COVID-19.

**Figure 2 life-12-00066-f002:**
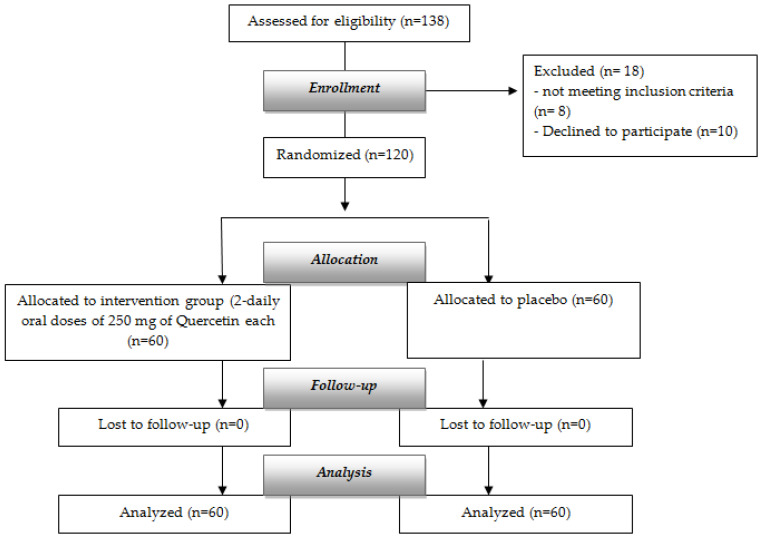
Flowchart of the study.

**Figure 3 life-12-00066-f003:**
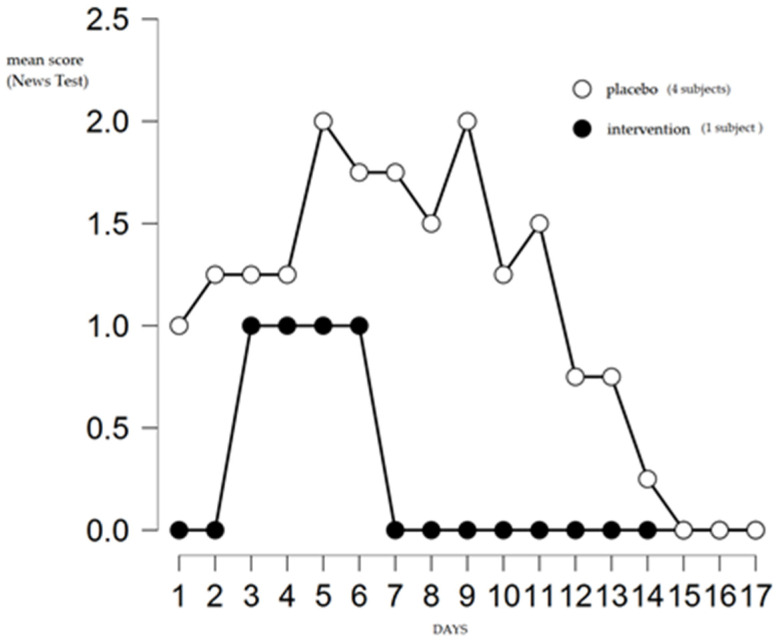
Symptoms by severity level assessed by NEWS in COVID-19-infected subjects.

**Figure 4 life-12-00066-f004:**
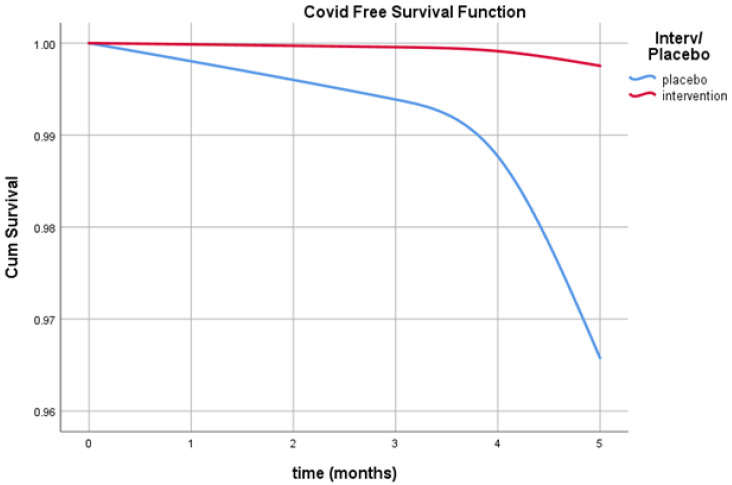
Survival without COVID-19 during follow-up time between groups.

**Table 1 life-12-00066-t001:** Baseline anthropometric characteristics of patients.

Variable	Placebo Group (N = 60)	Quercetin Group (N = 60)	Total (N = 120)
**Age** (**years**)	47.70 ± 13.67	50.88 ± 12.08	49.29 ± 12.94
**Height** (**m**)	1.74 ± 0.10	1.69 ± 0.10	1.71 ± 0.11
**Weight** (**kg**)	75.39 ± 12.02	69.19 ± 11.53	72.29 ± 12.13
**BMI** (**kg/m^2^**)	24.88 ± 2.74	24.10 ± 2.95	24.49 ± 2.86

**Table 2 life-12-00066-t002:** Overall sample demographic characteristics of subjects at baseline.

Variable	Number (Percentage)
Gender	
Male	63 (52.50%)
Female	57 (47.50%)
Smoker	
Yes	21 (17.50%)
No	99 (82.50%)
Physical activity	
Yes	44 (36.67%)
No	76 (63.33%)
Profession	
Job in contact with other people	77 (64.17%)
Job not in contact with other people	43 (35.83%)
Degree of education	
Middle-school diploma	19 (15.83%)
High School diploma	56 (46.67%)
University degree	45 (37.50%)

**Table 3 life-12-00066-t003:** Healing status at 10 and 10 + 7 days from baseline swab test.

Swab after 10 days	Placebo Group	Quercetin Group	Total
Negative	0	1	1
Positive	4	0	4
Total	4	1	5
**Swab after 10 + 7 days**			
Negative	4	0	4
Total	4	0	4

**Table 4 life-12-00066-t004:** Cox proportional-risk model.

	B	SE	Wald	Sig.	Exp(B)	95% CI for Exp(B)	
						Lower	Upper
**/PLACEBO/INTEGR**	2.646	1.304	4.117	**0.042 ***	14.097	1.094	181.596
**BMI**	−0.214	0.205	1.083	0.298	0.808	0.540	1.208
**Smoker**	2.447	1.260	3.771	0.052	11.552	0.978	136.514
**Physical activity**	0.617	1.012	0.372	0.542	1.854	0.255	13.468
**Profession**	−0.565	1.210	0.218	0.640	0.568	0.053	6.090
**Education**	2.068	1.094	3.570	0.059	7.908	0.926	67.559
**Age**	0.099	0.059	2.843	0.092	1.104	0.984	1.239

* *p* value in bold < 0.05.

## Data Availability

The data presented in this study are available in the main text.

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
