# Peer review of "Promising Effects of 3-Month Period of Quercetin Phytosome® Supplementation in the Prevention of Symptomatic COVID-19 Disease in Healthcare Workers: A Pilot Study"

_life, 2022, doi:10.3390/life12010066_

Round 1

Reviewer 1 Report

Overall manuscript presents a single centre, prospective randomized controlled study, where quercetine preparation, namely Quercetin Phytosome (250 mg 23 twice a day) was used to prevent from COVID-19 infection. Despite thorough preparation and conduct there are serios limitations and flaws in the manuscript that do not allow me to recommend it for publication in current form.

Major problems:

  1. Design of the study is not optimal and it is 3 months only study. Observation period is too short and number of events is too low to make any meaningful conclusions. It is not event-driven study and it is not powered to demonstrate the difference between the study arms. I calculated Chi-square test and difference between the groups is not significant (p=0.170532). Therefore, I strongly doubt that we can discuss this trial and Quercitin in terms of Efficacy. There is simply not enough evidence for that.
  2.  Authors should explain how they think Quercetin can be helpful to prevent disease contraction. As for the explanation about potential metabolic effects of Quercetin I would expect it to reduce manifestations of disease (as it was also demonstrated in the trial), but not contracting COVID-19. Comparing 1 positive case in Quercetine and 4 in Placebo is also unfair and conclusions cannot be drawn from 1 case with mild symptoms and there is high variability of covid-19 manisfestations in patients.
  3. There is no clear research hypothesis articulated. Any prospective trial should be hypothesis-driven in order to confirm or reject it. There might be many other factors that may have been contributing to the risk of contracting COVID-19. In lines 82-94 some evidence of Quercitine effects on virulence factors COVID-19 is mentioned, however, the information in this fragment is not fully consitent with cited references.
  4. It is unclear whether Quercetin reached suficient concentration in plasma or target organs to cause significant pharmacologic effect. Without this it is difficult to prove that the effect was caused by supplementation.
  5. I understand that it may be difficult to clearly identify exact mechanism of action of Quercitin in this case, but authors should have attepted to explain their findings in approriate way.

There are a number of minor issues as well, for example, limitations of the study are too superficial and do not address other important limitations, some of which were mentioned above.

Overall, my suggestion is to rewrite completely and resubmit the manuscript. Title should adequately reflect findings, it is a pilot study with limited power to demonstrate difference, so there is no way to demonstrate efficacy or effectiveness. Hypothesis should be clearly articulated and more details on how Quercetin may affect ability of virus to infect humans should be provided. I recommend to draw a scheme or a table on that. Clearly there are major limitation in the study, so this section shoul be carefully rewritten as well.

Reviewer 2 Report

this  ms "Effectiveness of 3-months period of quercetin phytosome sup-plementation in prevention of the risk of having COVID-19 infection in healthcare workers" is very interesting and merits publication in the journal.

However, some points must be emphasized

Quercetin Phytosome, it  must be fully explained what is

the role of antioxidant parameters must be included

Round 2

Reviewer 1 Report

The authors addressed major issues adequately. Now the paper is in acceptable form.